# Body mass index and risk of dying from a bloodstream infection: A Mendelian randomization study

Tormod Rogne[1,2,3]*, Erik Solligård[1,3], Stephen Burgess[4,5], Ben M. Brumpton[6,7,8], Julie Paulsen[9], Hallie C. Prescott[10,11], Randi M. Mohus[1,3], Lise T. Gustad[1,12], Arne Mehl[12], Bjørn O. Åsvold[6,13], Andrew T. DeWan[1,2], Jan K. Damås[1,14,15]‡

1 Gemini Center for Sepsis Research, Department of Circulation and Medical Imaging, NTNU Norwegian University of Science and Technology, Trondheim, Norway, 2 Department of Chronic Disease Epidemiology, Yale University School of Public Health, New Haven, Connecticut, United States of America, 3 Clinic of Anaesthesia and Intensive Care, St. Olavs Hospital, Trondheim University Hospital, Trondheim, Norway, 4 MRC Biostatistics Unit, University of Cambridge, Cambridge, United Kingdom, 5 Cardiovascular Epidemiology Unit, Department of Public Health and Primary Care, University of Cambridge, Cambridge, United Kingdom, 6 K.G. Jebsen Center for Genetic Epidemiology, Department of Public Health and Nursing, NTNU Norwegian University of Science and Technology, Trondheim, Norway, 7 MRC Integrative Epidemiology Unit, University of Bristol, Bristol, United Kingdom, 8 Clinic of Thoracic and Occupational Medicine, St. Olavs Hospital, Trondheim University Hospital, Trondheim, Norway, 9 Department of Medical Genetics, St. Olavs Hospital, Trondheim University Hospital, Trondheim, Norway, 10 Department of Medicine, University of Michigan, Ann Arbor, Michigan, United States of America, 11 VA Center for Clinical Management Research, Ann Arbor, Michigan, United States of America, 12 Department of Medicine, Levanger Hospital, Nord-Trøndelag Hospital Trust, Levanger, Norway, 13 Department of Endocrinology, St. Olavs Hospital, Trondheim University Hospital, Trondheim, Norway, 14 Centre of Molecular Inflammation Research, Department of Clinical and Molecular Medicine, NTNU Norwegian University of Science and Technology, Trondheim, Norway, 15 Department of Infectious Diseases, St. Olavs Hospital, Trondheim University Hospital, Trondheim, Norway

‡ These authors are joint senior authors on this work.
* tormod.rogne@yale.edu

**Data Availability Statement:** Data from the HUNT Study used in research projects available upon request to the HUNT Data Access Committee (hunt@medisin.ntnu.no) to research groups who

## Abstract

### Background

In observational studies of the general population, higher body mass index (BMI) has been associated with increased incidence of and mortality from bloodstream infection (BSI) and sepsis. On the other hand, higher BMI has been observed to be apparently protective among patients with infection and sepsis. We aimed to evaluate the causal association of BMI with risk of and mortality from BSI.

### Methods and findings

We used a population-based cohort in Norway followed from 1995 to 2017 (the Trøndelag Health Study [HUNT]), and carried out linear and nonlinear Mendelian randomization analyses. Among 55,908 participants, the mean age at enrollment was 48.3 years, 26,324 (47.1%) were men, and mean BMI was 26.3 kg/m². During a median 21 years of follow-up, 2,547 (4.6%) participants experienced a BSI, and 451 (0.8%) died from BSI. Compared with a genetically predicted BMI of 25 kg/m², a genetically predicted BMI of 30 kg/m² was associated with a hazard ratio for BSI incidence of 1.78 (95% CI: 1.40 to 2.27; *p* < 0.001) and for

meet the data availability requirements (described here: http://www.ntnu.edu/hunt/data). BMI GWAS source data (Yengo et al, 2018) are available from the GIANT Consortium (https://portals.broadinstitute.org/collaboration/giant/index.php/GIANT_consortium_data_files).

**Funding:** This study was in part funded by Samarbeidsorganet Helse Midt-Norge, NTNU, and The Research Council of Norway. TR was funded in part by a Fulbright Scholarship by the U.S-Norway Fulbright Foundation. BMB and BOÅ work in a research unit funded by Stiftelsen Kristian Gerhard Jebsen; Faculty of Medicine and Health Sciences, NTNU; The Liaison Committee for education, research and innovation in Central Norway; the Joint Research Committee between St. Olavs Hospital and the Faculty of Medicine and Health Sciences, NTNU; and the Medical Research Council Integrative Epidemiology Unit at the University of Bristol which is supported by the Medical Research Council and the University of Bristol [MC_UU_12013/1]. HCP is funded in part by K08 GM115859 from the US National Institutes of Health. The funding sources had no role in study design; in the collection, analysis, and interpretation of data; in the writing of the report; nor in the decision to submit the article for publication. The researchers were independent from the funders, and all authors had full access to all of the data in the study and can take responsibility for the integrity of the data and the accuracy of the data analysis.

**Competing interests:** I have read the journal's policy and the authors of this manuscript have the following competing interests: SB is a paid statistical reviewer for PLOS Medicine. HCP has current or prior grant funding from the NIH, AHRQ, and the US Department of Veterans Affairs. The other authors have declared that no competing interests exist.

**Abbreviations:** BMI, body mass index; BSI, bloodstream infection; GRS, genetic risk score; HR, hazard ratio; HUNT, Trøndelag Health Study; IPW, inverse probability weighting; IVW, inverse-variance-weighted; MR, Mendelian randomization.

BSI mortality of 2.56 (95% CI: 1.31 to 4.99; $p = 0.006$) in the general population, and a hazard ratio for BSI mortality of 2.34 (95% CI: 1.11 to 4.94; $p = 0.025$) in an inverse-probability-weighted analysis of patients with BSI. Limitations of this study include a risk of pleiotropic effects that may affect causal inference, and that only participants of European ancestry were considered.

## Conclusions

Supportive of a causal relationship, genetically predicted BMI was positively associated with BSI incidence and mortality in this cohort. Our findings contradict the "obesity paradox," where previous traditional epidemiological studies have found increased BMI to be apparently protective in terms of mortality for patients with BSI or sepsis.

## Author summary

### Why was this study done?

- It is well-recognized that overweight and obesity are associated with increased risk of bloodstream infection (BSI) and sepsis, but it is not fully understood whether this is due to body weight in itself or factors related to body weight (such as exercise or smoking habits).

- While a large number of studies have observed that BSI or sepsis patients who are overweight or obese have a *reduced* risk of dying from those diseases, there is reason to suspect that these findings are biased.

- We wanted to evaluate whether genetically predicted body mass index (BMI)—which is independent of lifestyle factors—was associated with risk of developing and dying from a BSI.

### What did the researchers do and find?

- We used clinical and genetic information from the Trøndelag Health Study in Norway on 55,908 participants representative of the adult Norwegian population.

- Similar to what has been found in non-genetic studies, we found that increased genetically predicted BMI was associated with an increased risk of developing a BSI.

- Contrary to many observational studies, we found that among BSI patients, being overweight or obese was associated with an increased risk of death from bloodstream infection.

### What do these findings mean?

- The findings of many previous observational studies of an apparently protective effect of overweight or obesity among patients with BSI or sepsis may be affected by other factors, such as accompanying characteristics of overweight or obese individuals, or by who ends up participating in the studies.

- In this cohort, higher genetically predicted BMI was associated with an increased risk of developing and dying from a BSI, also among BSI patients, and our findings support the worldwide initiative to reduce the prevalence of overweight and obesity.

## Introduction

Bloodstream infection (BSI) is caused by bacteria entering the bloodstream as a severe complication of an infection, and may in turn lead to sepsis, representing a dysregulated immune response to infection resulting in organ dysfunction and high mortality rate [1]. BSI is a common cause of death globally [2–4]. It is therefore important to identify factors that may reduce the risk of developing and dying from BSI.

Obesity is increasingly common worldwide [5], and is associated with greater risk of a wide range of diseases and all-cause mortality [6]. Traditional observational studies of general populations have found body mass index (BMI) to be positively associated with risk of BSI or sepsis [7–10], and above-normal BMI to be associated with BSI or sepsis mortality [7,9,11]. However, studies restricted to patients with BSI or sepsis have observed a considerable *reduced* mortality risk with increasing BMI [12–17], including a systematic review from 2017 [18]. It has therefore been advocated that one should study in what way obesity is protective for sepsis mortality, and that this may help inform new therapeutic strategies [17,18]. This counterintuitive reduction in disease progression or mortality among overweight and obese patients with a particular disease has been termed the "obesity paradox" and has been observed for other illnesses such as type 2 diabetes [19].

Some argue that the paradox may be explained by systematic error, and there has been a call for more rigorous studies to establish causal relationships [20–22]. There are 3 main areas of concern: selection bias, reverse causation, and confounding. If obesity is associated with BSI risk, nonobese patients may have other characteristics that caused their BSI that in turn are more strongly associated with mortality. This selection bias may make obesity appear protective in studies of mortality rate among all patients with BSI or sepsis [23]. Reverse causation may arise if measured BMI is affected by BSI (e.g., dehydration). Importantly, there may be confounding from factors such as chronic diseases and smoking habits that affect both BMI and BSI mortality. Mendelian randomization (MR) studies mimic randomized trials using genetic data as instruments for exposures. MR leverages information on genetic variants that segregate randomly at conception. If people with a genetic risk of being overweight or obese also have an increased risk of developing and dying from BSI, a causal relationship between BMI and mortality from BSI is strengthened (i.e., the relationship is likely independent of confounders and not subject to reverse causation) [24].

The aim of this study was to assess the causal association between BMI and risk of and mortality from BSI. We sought to overcome the limitations of the observational studies mentioned above by conducting an MR study in a general population of approximately 56,000 participants in Norway with 23 years of follow-up.

## Methods

This study is reported according to the preprint guideline for reporting of MR studies, STROBE-MR (S1 Checklist) [25] and the STROBE guideline for cohort studies (S2 Checklist) [26]. The Trøndelag Health Study (HUNT) is a series of cross-sectional surveys carried out in Nord-Trøndelag County, Norway. The county consists of 130,000 inhabitants, and is

representative of the general adult Norwegian population in terms of morbidity, mortality, sources of income, and age distribution [27]. The present study was based on the HUNT2 survey conducted in 1995–1997, to which 93,865 people were invited, and 65,236 (69.5%) participated.

Background characteristics such as age, sex, smoking status (current, former, never), lifestyle factors, education level, activity level, anthropometric measures, and self-reported history of cancer were collected once for each participant in the HUNT2 survey. Height and weight were measured by trained staff, and BMI calculated as weight in kilograms divided by the squared height in meters.

The personal identification number of Norwegian citizens was used to link the study population to all prospectively recorded blood cultures at the 2 community hospitals in the catchment area (Levanger and Namsos Hospitals), as well as St. Olavs Hospital in Trondheim, which serves as a tertiary referral center. Data on blood cultures were available from 1 January 1995 (from 1 September 1999 in Namsos Hospital) through the end of 2017. Dates of death and emigration out of Nord-Trøndelag County were obtained from the Norwegian population registry.

BSI was defined as positive blood culture of pathogenic bacteria, excluding bacteria such as *Corynebacterium* species, *Propionibacterium* species, and coagulase-negative *Staphylococcus* species often associated with blood culture contamination [28]. BSI mortality was defined as death within 30 days of BSI diagnosis.

Participants were followed until death or emigration out of Nord-Trøndelag, or to the end of December 2017.

Details about genotyping and imputation are provided in S1 Text. After sample and variant quality control, imputation was completed for 61,412 patients, all of European ancestry.

The genetic risk score (GRS) for BMI was calculated based on 939 of 941 near-independent, single nucleotide polymorphisms (SNPs; $p < 1 \times 10^{-8}$) identified as related to BMI in a meta-analysis of approximately 700,000 individuals (rs498240 and rs3819299 did not pass imputation quality control) [29].

Among patients with genotype data, BMI was available for 55,944. One participant was excluded due to outlying weight/height ratio. Participants who had a registered BSI before participating in the HUNT2 survey were excluded ($n = 35$). The final study sample consisted of 55,908 participants.

No protocol was written for this particular study, but the aims of the study were formalized prior to the analyses being conducted. The only substantial changes to the analyses were the addition of 2-sample MR analyses as suggested by peer-reviewers.

## Statistical analysis

The a priori main outcome of this MR study was the association between genetically predicted BMI and BSI mortality in the general population, while secondary analyses were MR analyses of association between genetically predicted BMI and (1) BSI incidence, (2) BSI incidence and mortality stratified by sex, and (3) BSI mortality among patients with BSI. Unless stated otherwise, the results presented in this study are evaluating the effect of genetically predicted BMI (as opposed to observed BMI).

The GRS for BMI was calculated by use of the "—score" command in PLINK (version 1.9), weighted based on the effect estimates from the meta-analysis by Yengo et al. [29]. We estimated the association between the GRS and BMI by use of linear regression, and between the GRS and risk of BSI or BSI mortality by use of Cox proportional hazards regression. Both models accounted for age and sex.

As previous studies have suggested a nonlinear association between BMI and BSI incidence and mortality [9], we fitted a fractional polynomial model as described elsewhere [30] and in S1 Text. In analyses assuming a linear relationship between exposure and outcome, we used the 2-stage least squares method with sandwich estimator for the error terms in the second stage [31]. In the MR analysis of the effect of BMI on BSI mortality among patients with BSI, day 0 was set to day of BSI diagnosis, and participants were followed for 30 days. To account for potential selection bias, we weighted participants from the total study population by the inverse probability of developing BSI, calculated based on the GRS, BMI, age, sex, history of cancer, smoking status, physical activity, and education [32]. The weighting was trimmed at the 99th percentile. This analysis was compared to an MR analysis without inverse probability weighting (IPW) and a similar traditional multivariable analysis adjusting for age, sex, history of cancer, smoking status, physical activity, and education.

Three main assumptions must be met for an MR study to be valid: (1) There is an association between the genetic instrument and the exposure; (2) the genetic instrument is not associated with the outcome other than through the exposure; and (3) the genetic instrument is not associated with confounders of the exposure–outcome association [24]. We performed a range of sensitivity analyses: (1) an analysis of the linear association between the GRS and BMI; (2) an analysis of the association between quartiles of the GRS and predefined confounders (due to signs of pleiotropic effects on smoking habits, the main analysis was run among never-smokers and then among previous or current smokers); (3) an MR analysis of BSI mortality in the general population restricted to near-independent SNPs associated with the *FTO* gene (robustly associated with BMI) [33]; (4) MR-Egger regression (random effects), inverse-variance-weighted (IVW) regression (random effects), and weighted median estimator analysis [34], assuming a linear relationship between BMI and BSI mortality in the general population; (5) comparison of analyses from the previous point with the same analyses conducted in a 2-sample design, using SNP–exposure associations from Yengo et al. [29] and SNP–outcome associations from HUNT (methods described in S1 Text); (6) comparison of our main MR analysis of BSI mortality in the general population with comparable traditional Cox regression adjusting for age, sex, history of cancer, smoking, physical activity, and education; and (7) repetition of the main analysis after adjustment for 5 ancestry-informative principal components computed by use of LASER v2.04 to account for potential population stratification [35].

Signs of selection bias were explored by restricting the study population to participants with BSI. We compared values of confounders for individuals in the highest BMI quartile (Q4) to those for individuals in the lowest BMI quartile (Q1) among patients with BSI, and compared this ratio (Q4/Q1) or difference (Q4 − Q1) with the concordant comparison in the total study population.

Finally, we did an MR sensitivity analysis stratified on whether the BSI was due to gram-positive or gram-negative bacteria.

R (version 3.4.1; packages metafor, ggplot2, gtools, Hmisc, ipw, lm.beta, sandwich, survey, survival, timereg, and lmtest) and Stata/SE 15.1 (College Station, TX, US; package mrrobust) were used.

### Ethical approval

The Regional Committee for Medical Research, Health Region IV, in Norway approved HUNT, and this project is regulated in conjunction with Norwegian Social Science Data Services.

### Results

A total of 55,908 participants were followed for a median of 21 years, amounting to 995,474 person-years. There were 2,547 (4.6%) first occurrences of BSI, and 451 (0.8% of total, 17.7%

**Table 1. Background characteristics.**

| Characteristic | Total population (n = 55,908) | BSI incidence (n = 2,547) | BSI death (n = 451) |
|---|---|---|---|
| Age (years)§ | 48.3 (36.5–62.3) | 63.6 (52.9–71.4) | 67.3 (57.1–74.5) |
| Male sex* | 26,324 (47.1) | 1,345 (52.8) | 263 (58.3) |
| BMI (kg/m²)^ | 26.3 (4.1) | 27.7 (4.5) | 27.9 (4.8) |
| Median follow-up time (years)§ | 21.1 (17.1–21.8) | 13.8 (8.4–18.3) | 13.3 (7.7–17.9) |
| Self-reported cancer* | 1,955 (3.7) | 144 (6.2) | 24 (5.9) |
| Smoking* | | | |
| Never | 23,594 (43.0) | 876 (35.2) | 156 (35.6) |
| Previous | 15,133 (27.6) | 893 (35.8) | 164 (37.4) |
| Current | 16,117 (29.4) | 723 (29.0) | 118 (26.9) |
| Physical activity* | | | |
| None | 3,821 (7.6) | 243 (11.9) | 54 (15.4) |
| Slight | 15,662 (31.0) | 714 (34.9) | 117 (33.3) |
| Moderate | 17,167 (34.0) | 693 (33.9) | 116 (33.1) |
| High | 13,810 (27.4) | 397 (19.4) | 64 (18.2) |
| Education* | | | |
| ≤9 years | 19,033 (35.7) | 1,305 (55.8) | 240 (58.8) |
| 10–12 years | 23,468 (44.0) | 762 (32.6) | 125 (30.6) |
| ≥13 years | 10,832 (20.3) | 274 (11.7) | 43 (10.5) |

BMI, body mass index; BSI, bloodstream infection. Data are presented as

^mean (standard deviation)

§median (25th–75th percentiles), or

*n (%). BSI incidence is based on first occurrence; otherwise, last occurrence is used. Education defined as follows: ≤9 years ("primary school 7–10 years, continuation school, folk high school"), 10–12 years ("high school, intermediate school, vocational school, 1–2 years high school" and "university qualifying examination, junior college, A levels"), and ≥13 years ("university or other post-secondary education, less than 4 years" and "university/college 4 years or more"). Activity defined as follows: none ("no light or vigorous activity"), slight ("<3 h light activity/week and no vigorous activity"), moderate ("≥3 h light activity/week or <1 h vigorous activity/week"), or high ("≥1 h vigorous activity/week").

of BSIs) deaths attributed to BSI. Participants who contracted a BSI or died from BSI were older, were more likely to be male and to be smokers, were less physically active and less educated, and had higher prevalence of self-reported history of cancer at study enrollment (Table 1).

The GRS explained 4.2% of the variation of BMI in our population (F-statistic = 2,461). There was a strong positive association between genetically predicted BMI and BSI incidence and mortality in the general population (Figs 1 and 2). A genetically predicted BMI of 20 kg/m², 30 kg/m², and 35 kg/m²—compared with 25 kg/m²—was associated with a hazard ratio (HR) for BSI incidence of 0.99 (95% confidence interval [CI]: 0.69 to 1.42; $p = 0.960$), 1.78 (95% CI: 1.40 to 2.27; $p < 0.001$), and 3.60 (95% CI: 2.18 to 5.93; $p < 0.001$), and for BSI mortality of 0.68 (95% CI: 0.48 to 0.96; $p = 0.029$), 2.56 (95% CI: 1.31 to 4.99; $p = 0.006$), and 8.57 (95% CI: 2.26 to 32.53; $p = 0.002$), respectively. Fig 1 and Fig 2 show tendencies of J-shaped associations, but are also compatible with linear relationships. A 1-kg/m² increase in genetically predicted BMI increased the risk of contracting BSI by 10% (95% CI: 5% to 16%; $p < 0.001$) and the risk of dying from BSI by 19% (95% CI: 6% to 33%; $p = 0.003$).

In the IPW analysis restricted to patients with BSI, we observed that a 1-kg/m² increase in genetically predicted BMI was associated with a non-statistically significant increased risk of BSI mortality of 11% (95% CI: −1% to 24%; $p = 0.069$). Genetically predicted BMI of 30 kg/m²,

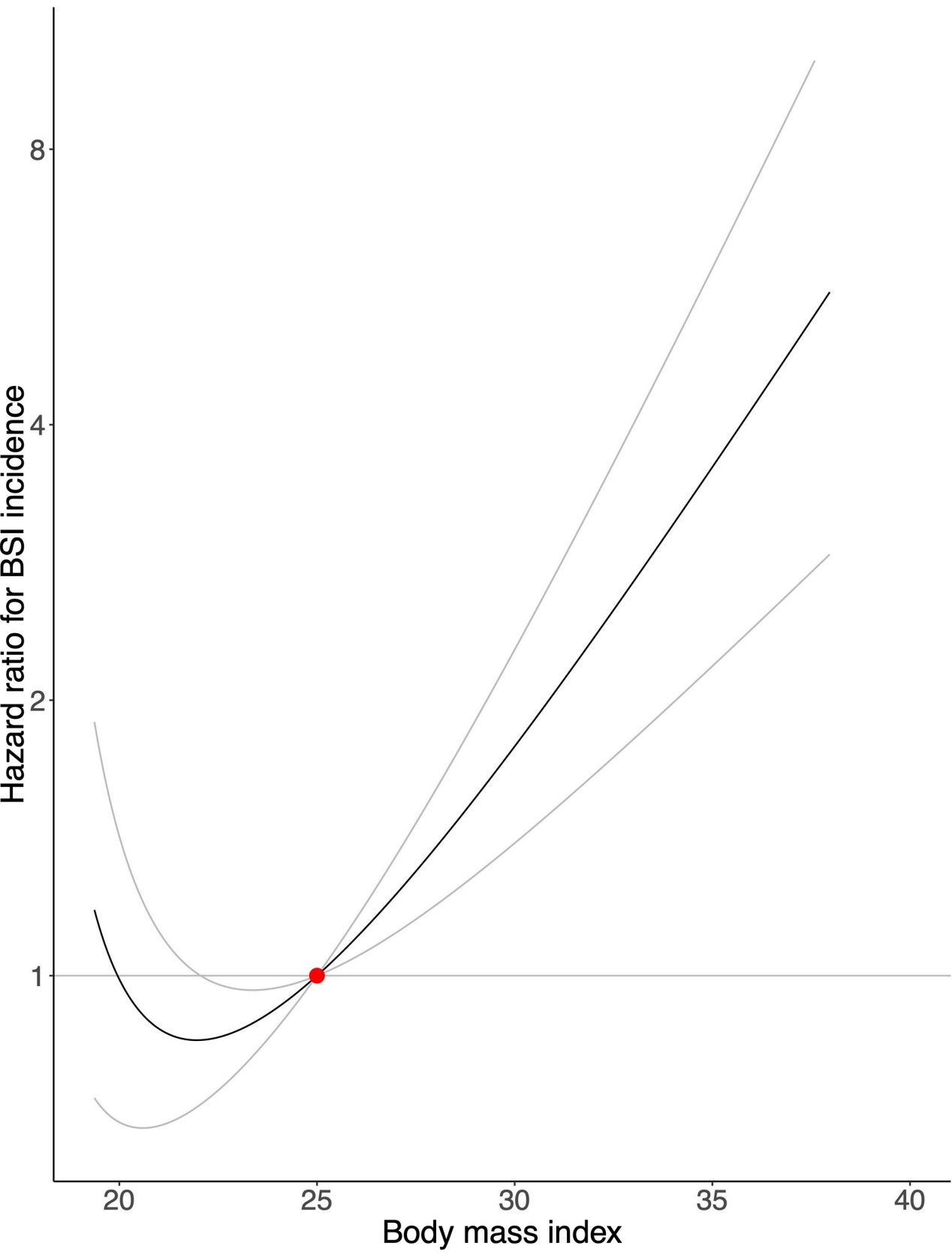

**Fig 1. Mendelian randomization analysis of body mass index and bloodstream infection incidence.** The association between genetically predicted body mass index and risk of contracting a bloodstream infection (BSI), with a body mass index of 25 kg/m$^2$ as reference (red dot). Gray lines represent 95% confidence intervals.

compared with 25 kg/m$^2$, was associated with a HR for mortality of 2.34 (95% CI: 1.11 to 4.94; $p = 0.025$) (Fig 3). For comparison, a 1-kg/m$^2$ increase in BMI in the non-IPW MR analysis was associated with an increase in BSI mortality risk of 12% (95% CI: 1% to 25%; $p = 0.039$), and in the non-IPW traditional multivariable Cox regression model, it was associated with a non-statistically significant increase in mortality risk of 2% (95% CI: −1% to 5%; $p = 0.188$).

The sex-stratified associations between BMI and BSI incidence and mortality in the general population are presented in S1 and S2 Figs, respectively.

## Sensitivity analyses

The traditional Cox regression model provided a weaker association between BMI and BSI incidence and mortality in the general population compared with the MR analysis (S1 and S2 Tables).

The results were unchanged after adjustment for 5 ancestry-informative principal components; in other words, there were no signs of population stratification (S3 Fig). When using only SNPs associated with the *FTO* gene to calculate the GRS, which explained 0.4% of BMI variation (*F*-statistic = 218), the results were consistent, though with wider CIs (S4 Fig).

Potential confounders were tabulated against BMI quartiles (S3 Table) and GRS quartiles (S4 Table). BMI was positively associated with self-reported history of cancer, never-smoking status, and age, and negatively associated with current smoking status, education, and physical activity. For the GRS, there was a positive association with current smoking status, while the other confounders were minimally associated ($R^2 < 0.1\%$). Sensitivity analyses of never-smokers and ever-smokers yielded the same association between BMI and BSI mortality as in the main analysis (S5 Fig).

MR-Egger regression supported a causal association between BMI and BSI mortality in the general population adjusted for directional pleiotropy (HR = 1.18; 95% CI: 1.04 to 1.33; $p = 0.011$), and the average pleiotropic effect of the SNPs was null (HR = 1.00; 95% CI: 0.99 to 1.00; $p = 0.476$) (S5 Table; S6 Fig) [34]. We observed similar findings using the IVW method and weighted median estimator ($p = 0.002$ and $p = 0.081$, respectively). The comparable analyses in a 2-sample design yielded a similar association between BMI and risk of dying from BSI in the general population: The odds ratios in MR-Egger regression, IVW regression, and weighted median estimator analysis for a one standard deviation increase of BMI were 1.98 (95% CI: 0.95 to 4.18; $p = 0.070$), 1.89 (95% CI: 1.33 to 2.67; $p < 0.001$), and 2.09 (95% CI: 1.10 to 3.97; $p = 0.025$), respectively (S5 Table).

Evaluating signs of selection bias, we found that BSI patients in the highest BMI quartile, compared with those in the lowest quartile, were less likely to report a history of cancer, which was not the case in the general population (S6 Table). When we examined participants who would eventually develop BSI, participants in the highest BMI quartile—relative to those in the lowest BMI quartile—had a lower prevalence of risk factors and a higher prevalence of protective factors, compared with the concordant prevalences in the general population.

Among the underweight and normal weight participants (BMI < 25 kg/m$^2$), genetically predicted BMI was more strongly associated with BSI mortality in the general population for gram-positive infections than gram-negative infections (S7 Fig), but there were no differences among the overweight and obese participants.

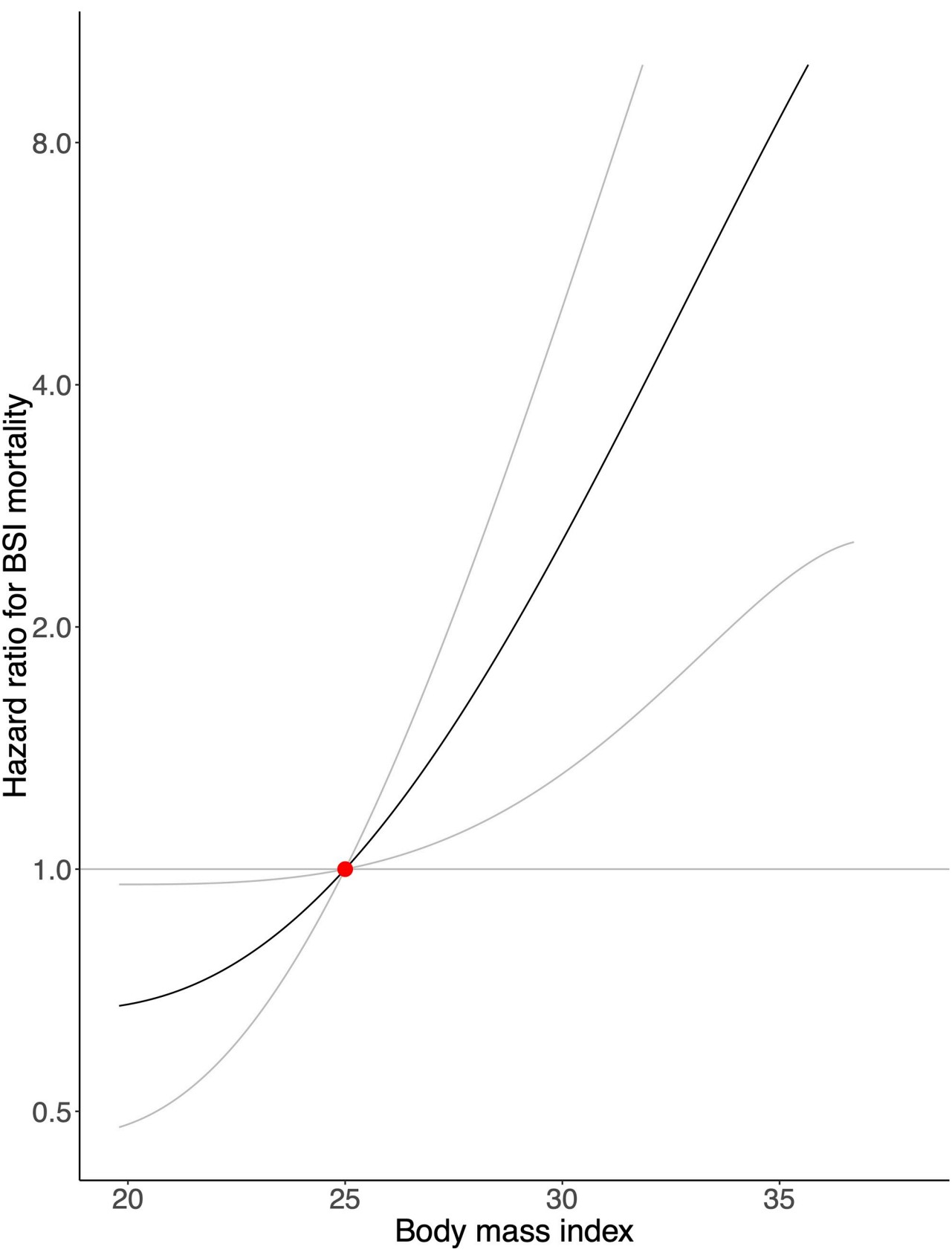

**Fig 2. Mendelian randomization analysis of body mass index and bloodstream infection mortality in the general population.** The association between genetically predicted body mass index and risk of dying from a bloodstream infection (BSI) in the general population, with a body mass index of 25 kg/m$^2$ as reference (red dot). Gray lines represent 95% confidence intervals.

## Discussion

This MR study shows a positive association between genetically predicted BMI and BSI mortality, both in the general population and among patients with BSI, opposing the obesity paradox. Similarly, genetically predicted BMI was positively associated with BSI incidence.

In several studies, overweight and obese participants have been found to be at an increased risk of developing sepsis or BSI [7–10]. One of these studies also performed MR analyses of BMI and population risk of developing sepsis, but found no association, which may be due to the use of a weak genetic instrument [10]. Three recent population-based studies found overweight [7] and obese [7,9] participants to be at increased risk of BSI mortality, or observed a positive association between BMI and BSI mortality [11]. However, these observational studies may be affected by reverse causation and residual confounding.

The present study used a similar study population to that of Paulsen et al. [9], but had 6 more years of follow-up, 9,000 fewer participants (due to dependence on genotyping), and 55 (14%) more deaths due to BSI [9]. While Paulsen et al. found that BSI mortality risk started to increase at around BMI 30 kg/m$^2$, we found it to start to increase at around 20 kg/m$^2$. This observed difference is likely because the MR design has accounted for residual negative confounding. Our study found a stronger association between BMI and BSI mortality than the 2 other population-based studies [7,11]. MR estimates represent the impact of long-term differences in the trajectory of a risk factor over the life course; this impact is likely to differ from the impact of short-term interventions on the risk factor. Additionally, with a binary outcome, the estimates are marginal across covariates, and hence will differ from estimates from a multivariable-adjusted regression [36].

Our findings are in line with experimental animal studies of obesity and sepsis mortality [37]. However, human observational studies restricted to patients with sepsis or BSI have consistently found that there is a *reduced* risk of sepsis or BSI mortality among overweight and obese patients [18]. A systematic review from 2017 found markedly reduced mortality from sepsis among overweight and obese patients compared with those of normal weight [18]. Other hospital-based studies have reported similar findings [12–15], including a recent study of approximately 55,000 patients from 139 hospitals in the United States [17].

However, all of these hospital-based studies are likely affected by selection bias, in addition to possible reverse causation and confounding [38,39]. Similarly, when we restricted our study population to those that would develop BSI, the risk of BSI mortality among participants in the highest BMI quartile was reduced. Adjustment for introduced selection bias will often be insufficient [40].

One way to reduce selection bias is to use IPW [32]. In our IPW analysis of BSI patients, we observed a positive association between genetically predicted BMI and mortality similar to that of the total study population, albeit somewhat weaker. The finding of the comparable unweighted MR analysis was similar to that of the weighted one, which is supported by simulations that show that moderate selection bias will only lead to slight bias in a typical MR analysis [32]. Additionally, we observed a stronger association between BMI (both genetically predicted and observed) and BSI mortality in the general population than between BMI and BSI incidence, which would be expected if BMI increases BSI mortality risk among BSI patients.

In terms of the MR assumptions, assumption 1 was met as there was a strong association between the GRS and the exposure. Smoking was unevenly distributed at different levels of the

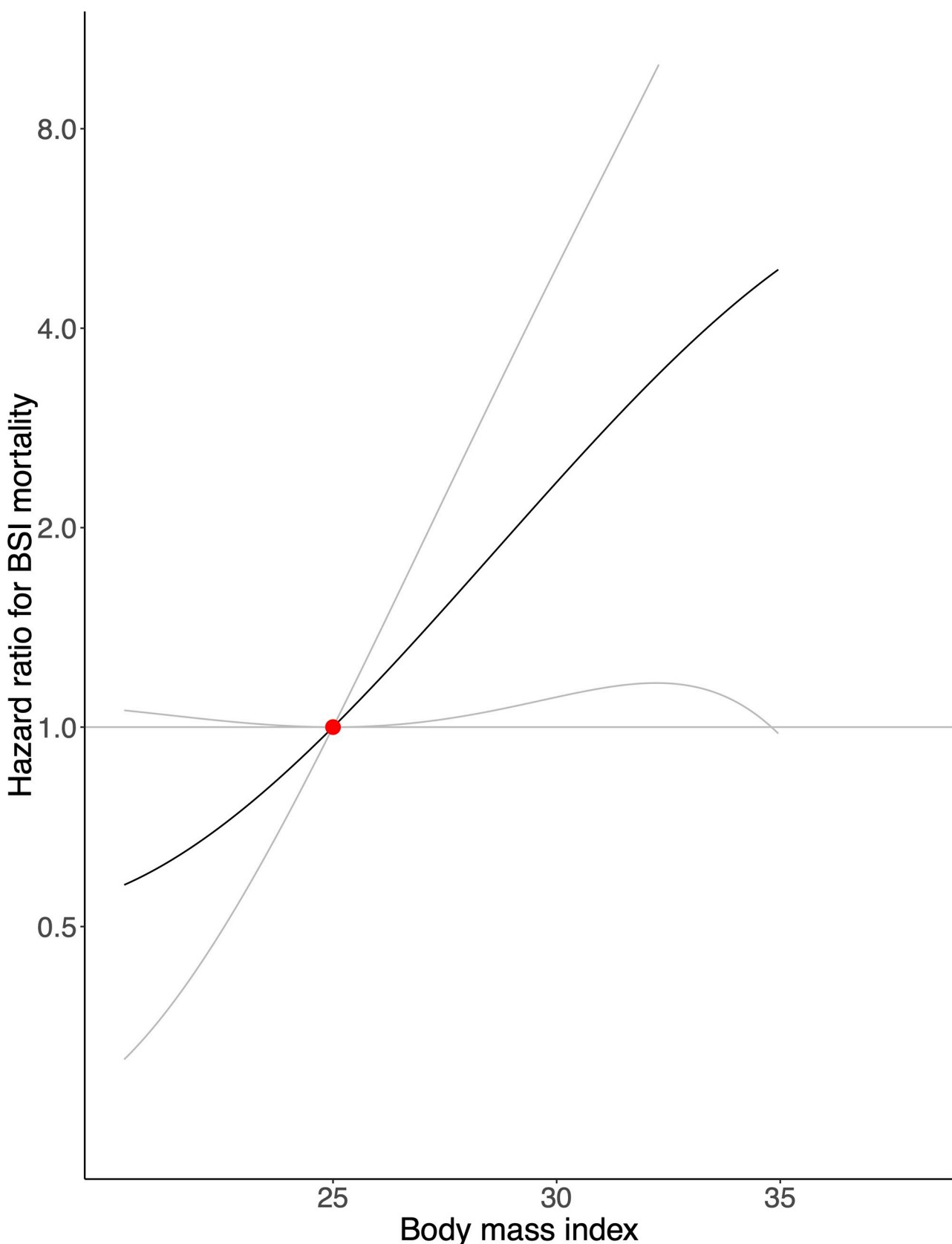

**Fig 3. Mendelian randomization analysis of body mass index and bloodstream infection mortality among patients with bloodstream infection.**
The association between genetically predicted body mass index and risk of dying from a bloodstream infection (BSI) among patients with a

bloodstream infection, with a body mass index of 25 kg/m$^2$ as reference (red dot). The analysis was weighted for the inverse probability of contracting a bloodstream infection. Gray lines represent 95% confidence intervals.

GRS, potentially violating assumption 3. However, in analyses stratified by smoking status, we observed the same finding as in our main analysis. Additionally, assumption 2 was strengthened by MR-Egger regression, IVW regression, median estimator analysis, and the analysis restricted to the *FTO* gene as instrument. In the 2-sample sensitivity analysis, the linear association between BMI and risk of dying from a BSI in the general population was similar, albeit with less precision, compared with the 1-sample analyses. The larger standard errors suggest heterogeneity between the populations used by Yengo et al. [29] and in this study.

## Strengths and limitations

To our knowledge, this is the first study to date to evaluate the association between BMI and risk of dying from BSI or sepsis using an MR design. This allows us to assess an association that is much less susceptible to reverse causation and unmeasured confounding than traditional observational studies. The MR design and IPW based on pre-disease background characteristics in the analysis of BSI mortality among BSI patients reduces the risk of selection bias. The genetic instrument for BMI that we used has been successfully applied in other recently published MR studies [41,42]. Importantly, although it is recommended that researchers create the genetic instrument based on the full set of SNPs associated with the desired exposure, this may increase the risk of pleiotropic effects and affect causal inference [24]. We therefore conducted a wide range of suggested sensitivity analyses [24], which supported that our findings were not meaningfully affected by pleiotropy. MR-Egger regression may be biased and affected by residual confounding when applied in a 1-sample setting [43]; however, in our 2-sample analyses we observed consistent results across the different sensitivity analyses. As the instruments were selected from an independent sample, our MR analyses will tend not to be affected by winner's curse [44]. We assumed the phenotype–disease association to be log-linear, which may bias the instrumental variable estimate depending on the properties of unmeasured confounders [45]; the non-instrumental variable analyses yielded similar but weaker associations. Given that we used a homogenous population of participants of northern European ancestry, the risk of population stratification is reduced, and it increases the precision of the GRS, which was based on a genome-wide association study of individuals of similar ancestry [29]. We encourage replication of our study in non-European cohorts to evaluate the generalizability of our findings to other populations. The vast majority of patients with BSI have sepsis, but not all patients with sepsis will have a positive blood culture [46]. Through record linkage with all hospitals in the region of interest, we had complete information on all relevant BSI occurrences. However, while blood culture tests are the gold standard to diagnose BSI, the sensitivity ranges between 10% and 50%, which may have reduced the statistical power in our study [47]. We overcame the issue of power by following approximately 56,000 participants representative of the adult Norwegian population for 23 years. Finally, we were able to allow for potential nonlinear associations between the exposure and the outcome, which has previously been challenging in MR studies [30].

## Conclusion

In this Norwegian population-based cohort using an MR design, we found that any increase in genetically predicted BMI increased the risk of both BSI incidence and BSI mortality in the general population. Importantly, in an IPW MR analysis additionally accounting for selection bias, we found that increased genetically predicted BMI increased the risk of mortality also

among patients with BSI, opposing the obesity paradox. While other researchers have suggested that the "protective" effect of high BMI among patients with severe infectious diseases may help inform new therapeutic strategies [17,18], our findings do not encourage such initiatives. Given the steady, worldwide increase of mean BMI among adults, and that BSI is a common cause of death, these findings strongly support interventions to reduce the prevalence of overweight and obesity [2,5]. The efficacy of such interventions on the incidence of and mortality from BSI should then be studied.

## Supporting information

**S1 Checklist. STROBE-MR checklist for Mendelian randomization studies.**
(PDF)

**S2 Checklist. STROBE checklist for cohort studies.**
(PDF)

**S1 Fig. Mendelian randomization analysis of body mass index and bloodstream infection incidence, stratified by sex.** The association between genetically predicted body mass index and risk of contracting a bloodstream infection among men (left panel) and women (right panel), with a body mass index of 25 kg/m$^2$ as reference. Gray lines represent 95% confidence intervals.
(PDF)

**S2 Fig. Mendelian randomization analysis of body mass index and bloodstream infection mortality in the general population, stratified by sex.** The association between genetically predicted body mass index and risk of dying from a bloodstream infection among men (left panel) and women (right panel) in the general population, with a body mass index of 25 kg/m$^2$ as reference. Gray lines represent 95% confidence intervals.
(PDF)

**S3 Fig. Mendelian randomization analysis of body mass index and bloodstream infection mortality in the general population, adjusted for 5 principal components.** The association between genetically predicted body mass index and risk of dying from a bloodstream infection in the general population, with a body mass index of 25 kg/m$^2$ as reference, adjusting for 5 principal components. Gray lines represent 95% confidence intervals.
(PDF)

**S4 Fig. Mendelian randomization analysis of body mass index and bloodstream infection mortality in the general population, with an instrument based on the *FTO* gene.** The association between genetically predicted body mass index and risk of dying from a bloodstream infection in the general population, with a body mass index of 25 kg/m$^2$ as reference. Gray lines represent 95% confidence intervals. Single nucleotide polymorphisms used were rs11075986, rs16952479, rs2075205, rs3751813, rs8047395, rs9922708, and rs9931164.
(PDF)

**S5 Fig. Mendelian randomization analysis of body mass index and bloodstream infection mortality in the general population, stratified by smoking status.** The association between genetically predicted body mass index and risk of dying from a bloodstream infection among never-smokers (left panel) and ever-smokers (right panel) in the general population, with a body mass index of 25 kg/m$^2$ as reference. Gray lines represent 95% confidence intervals.
(PDF)

**S6 Fig. MR-Egger plot of bloodstream infection mortality in the general population.** Single nucleotide polymorphism–body mass index association and single nucleotide polymorphism–bloodstream infection mortality association in the general population, with inverse-variance-weighted regression for comparison.
(JPG)

**S7 Fig. Mendelian randomization analysis of body mass index and bloodstream infection mortality in the general population, stratified by gram-positive and gram-negative bacteria.** The association between genetically predicted body mass index and risk of dying from a bloodstream infection due to gram-positive (left panel) and gram-negative bacteria (right panel), with a body mass index of 25 kg/m$^2$ as reference. Gray lines represent 95% confidence intervals. There were 220 deaths due to gram-positive bloodstream infection and 215 deaths due to gram-negative bloodstream infection.
(PDF)

**S1 Table. Two-stage least-squares Mendelian randomization versus multivariable Cox regression of body mass index and bloodstream infection incidence.** BMI, body mass index; BSI, bloodstream infection; HR, hazard ratio. The 6 strata correspond to the BMI categories underweight, normal weight, overweight, obese class I, obese class II, and obese class III (from the top). The BMI strata were created based on residual BMI and observed BMI in the Mendelian randomization and multivariable analysis, respectively. The HR corresponds to a 1-kg/m$^2$ increase in BMI within that stratum.
(DOCX)

**S2 Table. Two-stage least-squares Mendelian randomization versus multivariable Cox regression of body mass index and bloodstream infection mortality in the general population.** BMI, body mass index; BSI, bloodstream infection; HR, hazard ratio; INF, infinity. The 6 strata correspond to the BMI categories underweight, normal weight, overweight, obese class I, obese class II, and obese class III (from the top). The BMI strata were created based on residual BMI and observed BMI in the Mendelian randomization and multivariable analysis, respectively. The HR corresponds to a 1-kg/m$^2$ increase in BMI within that stratum.
(DOCX)

**S3 Table. Distribution of potential confounders by body mass index quartiles.** BMI, body mass index; Q, quartile; SD, standard deviation. Post-secondary defined as at least some university or other post-secondary education. Moderate/high activity defined as ≥3 h light activity/week or any vigorous activity/week.
(DOCX)

**S4 Table. Distribution of potential confounders by genetic risk score for body mass index quartiles.** BMI, body mass index; GRS, genetic risk score; Q, quartile; SD, standard deviation. Post-secondary defined as at least some university or other post-secondary education. Moderate/high activity defined as ≥3 h light activity/week or any vigorous activity/week.
(DOCX)

**S5 Table. Mendelian randomization sensitivity analyses of linear association between body mass index and bloodstream infection mortality in the general population.** HR, hazard ratio; IVW, inverse-variance-weighted; OR, odds ratio. The analyses assume a linear relationship between body mass index and bloodstream infection mortality in the general population using the same 939 single nucleotide polymorphisms (SNPs) as used to create the genetic risk score. Two-sample analyses use SNP–exposure associations from Yengo et al. [29] and SNP–outcome associations from HUNT. The $I^2$ values of the SNP–exposure associations were 54%

in the 1-sample MR-Egger regression and 92% in the 2-sample MR-Egger regression. Effect estimates reported as HR for 1 unit increase of body mass index in 1-sample analyses and as OR for 1 standard deviation increase of body mass index in 2-sample analyses.
(DOCX)

**S6 Table. Distribution of potential confounders by body mass index quartiles among patients with bloodstream infection.** BMI, body mass index; BSI, bloodstream infection; Q, quartile; SD, standard deviation. The final column, for dichotomous covariates, compares the Q4/Q1 ratio among patients with BSI with the Q4/Q1 ratio in the general population (S3 Table), and similarly compares the mean difference of Q4 − Q1 among BSI participants with Q4 − Q1 in the general population. Post-secondary defined as at least some university or other post-secondary education. Moderate/high activity defined as ≥3 h light activity/week or any vigorous activity/week.
(DOCX)

**S1 Text. Supplementary description of methods.**
(DOCX)

## Acknowledgments

HUNT is a collaboration between the HUNT Research Centre (Faculty of Medicine and Health Sciences, Norwegian University of Science and Technology), Nord-Trøndelag County Council, Central Norway Regional Health Authority, and Norwegian Institute of Public Health.

This paper does not necessarily reflect the position or policy of the US Government or Department of Veterans Affairs.

## Author Contributions

**Conceptualization:** Tormod Rogne, Erik Solligård, Stephen Burgess, Bjørn O. Åsvold, Andrew T. DeWan, Jan K. Damås.

**Data curation:** Tormod Rogne, Erik Solligård, Ben M. Brumpton, Julie Paulsen, Randi M. Mohus, Arne Mehl, Jan K. Damås.

**Formal analysis:** Tormod Rogne, Stephen Burgess, Ben M. Brumpton, Bjørn O. Åsvold.

**Funding acquisition:** Erik Solligård, Andrew T. DeWan, Jan K. Damås.

**Investigation:** Stephen Burgess, Randi M. Mohus, Bjørn O. Åsvold.

**Methodology:** Tormod Rogne, Stephen Burgess, Ben M. Brumpton, Bjørn O. Åsvold, Andrew T. DeWan.

**Project administration:** Erik Solligård, Ben M. Brumpton, Lise T. Gustad, Arne Mehl, Andrew T. DeWan, Jan K. Damås.

**Resources:** Erik Solligård, Andrew T. DeWan, Jan K. Damås.

**Software:** Erik Solligård.

**Supervision:** Erik Solligård, Stephen Burgess, Andrew T. DeWan, Jan K. Damås.

**Validation:** Erik Solligård, Hallie C. Prescott.

**Visualization:** Stephen Burgess.

**Writing – original draft:** Tormod Rogne, Erik Solligård, Stephen Burgess, Bjørn O. Åsvold, Andrew T. DeWan, Jan K. Damås.

**Writing – review & editing:** Tormod Rogne, Erik Solligård, Stephen Burgess, Ben M. Brumpton, Julie Paulsen, Hallie C. Prescott, Randi M. Mohus, Lise T. Gustad, Arne Mehl, Bjørn O. Åsvold, Andrew T. DeWan, Jan K. Damås.

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
