## [Editor Report · Decision Letter 0]

7 Jan 2020

Dear Dr Rogne, 

Thank you for submitting your manuscript entitled "Body Mass Index and Risk of Dying from a Bloodstream Infection: A Mendelian Randomization Study of 56,000 Subjects from the HUNT Study With 23 Years Follow-Up" for consideration by PLOS Medicine.

Your manuscript has now been evaluated by the PLOS Medicine editorial staff and I am writing to let you know that we would like to send your submission out for external peer review.

Please re-submit your manuscript within two working days, i.e. by Jan 9 2020 11:59PM.

**Please be aware that, due to the voluntary nature of our reviewers and academic editors, manuscript assessment may be subject to delays during the holiday season. Thank you for your patience.**

Kind regards,

Louise Gaynor-Brook, MBBS PhD,

PLOS Medicine

---

## [Decision Letter · Decision Letter 1]

23 Apr 2020

Dear Dr. Rogne,

Thank you very much for submitting your manuscript "Body Mass Index and Risk of Dying from a Bloodstream Infection: A Mendelian Randomization Study of 56,000 Subjects from the HUNT Study With 23 Years Follow-Up" (PMEDICINE-D-19-04607R1) for consideration at PLOS Medicine. 

[LINK]

In light of these reviews, I am afraid that we will not be able to accept the manuscript for publication in the journal in its current form, but we would like to consider a revised version that addresses the reviewers' and editors' comments. Obviously we cannot make any decision about publication until we have seen the revised manuscript and your response, and we plan to seek re-review by one or more of the reviewers. 

We expect to receive your revised manuscript by Jun 15 2020 11:59PM. Please email us (plosmedicine@plos.org) if you have any questions or concerns.

We look forward to receiving your revised manuscript. 

Sincerely,

Emma Veitch, PhD

PLOS Medicine

On behalf of Clare Stone, PhD, Acting Chief Editor,

PLOS Medicine

plosmedicine.org

*Please structure your abstract using the PLOS Medicine headings (Background, Methods and Findings, Conclusions) - the Methods and Findings section should be one single section (“Methods and findings”).

*In the last sentence of the Abstract Methods and Findings section, please describe the main limitation(s) of the study's methodology.

*At this stage, we ask that you include a short, non-technical Author Summary of your research to make findings accessible to a wide audience that includes both scientists and non-scientists. The Author Summary should immediately follow the Abstract in your revised manuscript. This text is subject to editorial change and should be distinct from the scientific abstract. Please see our author guidelines for more information: https://journals.plos.org/plosmedicine/s/revising-your-manuscript#loc-author-summary

*Although in general the paper is very clearly reported, the authors might wish to know that there is a draft guideline, STROBE-MR, for reporting Mendelian Randomization studies (https://peerj.com/preprints/27857/) - and they may wish to use, and reference, this guideline to help enhance the depth of reporting in their paper, if appropriate. 

Comments from the reviewers:

Reviewer #1: This is an interesting study assessing the causal relationship between body mass index and bloodstream infection (risk and mortality) under a Mendelian Randomization framework. The authors used information on 55,908 participants of the HUNT study that were followed during a median of 21 years. In order to allow for non-linear associations, they also conducted a non-linear Mendelian randomization approach. They found that elevated genetically-predicted BMI was causally associated with increased risk and mortality of bloodstream infection. 

Given that this is the first study evaluating the causal association between body mass index and bloodstream infection in a Mendelian randomization framework, in addition to allowing for non-linear associations, these are interesting results that can help to understand the mechanisms of body mass index on health outcomes. 

I found the article well organized and clearly written. The main and sensitivity statistical analyses are appropriate, and the results and conclusions are generally well presented.

I only have two minor comments regarding the presentation of the results.

1. As in Supplemental Table 1, the authors should add a table with the results for the association between BMI and bloodstream incidence. These results could also be added in Supplemental Table 1 in new rows. 

2. In "Supplemental Table 3: Evaluation of MR-assumptions" the title is not clear. Are those the results from the robust MR methods (MR-Egger, IVW and weighted median) for the association between BMI and bloodstream infection mortality? Or incidence? Or it is another thing? Please clarify. In fact, it would be good to have a table with the overall results for bloodstream infection mortality and incidence with the distinct techniques used in this study: 2 stage least squares, MR-Egger, IVW, weighted median, etc. This table would strengthen the presentation of the results. 

Suggestion: I would combine Figure 4A and Figure 4B in a single figure with 2 panels. The same for Supplemental Figure 1A and 1B, and for Supplemental Figure 4A and 4B. By showing together the associations for men and women (never smoker and ever smoker) you will help the reader to see the comparisons more easily. 

Reviewer #2: In this manuscript, authors use a population-based cohort from Norway with 23 years follow up and evaluated the impact of BMI on the risk of bloodstream infection and associated mortality risk. They use Mendelian randomization in an attempt to minimize bias, confounding and reverse causation. I agree that associations in observational studies are subject to a number of biases and attempts to obtain answers using methodologies that mitigate bias, especially for clinically important questions such as this are warranted. However, I have a number of concerns that I think need to be addressed first. 

Major

1. It is unclear to me as to whether authors correlated genotype-predicted BMI with actual BMI. Please explain. Granted that the genetic predisposition for higher BMI is a nice plus to this study, however BMI is also very much impacted by environmental factors and as such, BMI changes considerably in sepsis/bacteremic patients on presentation especially when they receive significant fluid resuscitation. The time interval between BMI measurement and onset of BSI is very relevant. It is difficult to associate BMI with outcome if the two are very far apart in time. Please compare results of predicted BMI and actual BMI based associations. 

2. Authors had access to bloodstream pathogen species (to be able to exclude contaminant species). Please provide sensitivity analyses by organism type (gram positive ve negative), It would be valuable to know of there is pathogen-level heterogeneity of outcome. 

3. The first sentence of the abstract is long and confusing as written. Please have succinct statements in the abstract so you do not lose your audience at the outset. 

4. Some more detail on the way patients were followed up would be helpful. How often were they seem after the initial survey? How often were measurements llke BMI done. 

5. Authors appear to have aces to microbiology data from the hospitalizations at 2 community and 1 tertiary care hospital. WHat other variables did they have access to among hospitalized patients? 

6. Related to point #5, I think one of the biggest limitations of this study is the lack of physiologic and lab variables for risk adjustment of severity of acute illness. These data might be available from the hospital encounters. In line with authors attempts to minimize confounding, the elephant in the room is the absence of adjustment for baseline acute illness upon presentation with BSI. This really biases the outcome and put authors' findings and their claims into question.

7. The discussions section on "Strengths and Limitations" mostly displays strengths alone. Please report limitations as well. 

8. Authors need a stronger conclusion statement. Saying that this work promote efforts to encourage weight loss is not enough. 

Minor 

1. Lack of line numbers makes the review process a bit more cumbersome. 

2. On page 11, paragraph 2: what was the basis of choose 30 days? Any justification for the same? 

Reviewer #3: Thank you for asking me to review this article which concerns the relationship between obesity and blood stream infection (BSI). There remains considerable controversy over the relationship between sepsis mortality and obesity, with conflicting results from studies which have used a variety of methods to investigate the relationship. In this manuscript Rogne and colleagues use a Mendelian randomisation technique to examine the relationship between genetically predicted body-mass index and BSI incidence and mortality, demonstrating positive relationships between both these and genetically predicted BMI. As the first MR-based study to investigate the relationship between genetically predicted obesity and BSI incidence/mortality, this is a significant addition to the literature which overcomes several of the problems associated with conventional observational studies. As the authors correctly note, BSI and sepsis are not synonymous although most patients with BSI will meet the current criteria for sepsis and therefore these findings are likely to be of relevance beyond BSI and indicate methods for similar studies in other infectious disease syndromes.

This is a well conducted study with a rigorous approach to examining potential confounders, and a measured discussion of the findings with due note given to findings such as the potential J-shaped relationship without over-interpretation. My only comment is that although positive blood cultures (excluding common contaminants) is specific for blood stream infection, it is insensitive as it relies on the growth of bacterial organisms - which either organism fastidiousness or inter-current antibiotics may prevent. It is therefore possible, indeed likely, that some of the individual included in this study as 'non-BSI' patients will have had a missed BSI. Recent studies of molecular diagnostics, (e.g. Nguyen MH, et al. Annals of Internal Medicine 2019) suggest that BSI may be over 4 times more prevalent amongst hospitalised patients than is detected by growth-based methods. I suggest that the authors discuss this as a potential weakness in this study, and make it clear that their findings -at present- relate only to culture-positive BSI.

Reviewer #4: Body mass index (BMI) has been associated with the risk of mortality of bloodstream infection (BSI). But the direction of the effect is unclear, because BMI seems to be a risk factor in the general population and a protective factor in patience with BSI and sepsi.

The authors performed a one-sample Mendelian randomization (MR) analysis using the HUNT study. MR analyses were conducted with a weighted genetic risk score composed of 939 independent variants. That study support the hypothesis of BMI as a risk factor.

The paper is well written. The MR study is very well performed. My specific comments are detailed hereafter.

1. The authors should comment about the bias due to the binary outcome

2. However, as a strength they should also mention that those analyses are not biased by the winner course.

3. About the MR-Egger, they should comment something about the measurement error.

4. In my opinion, in order to evaluate the robustness of sensitivity analyses, they should comment about additional assumptions, which are made for each MR method. For example, what about the independence between GX and GY associations in the context of one-sample MR study.

5. They should comment on the use of two-sample MR methods using one sample. Maybe, it could be interesting to compare their results with a MR study performed using GX and GY estimates from two different data sources, which have to be homogeneous, of course.

6. Just a small comment, be consistent with the notation for confidence intervals, don't use the dash after 95% CI, but for example the colon.

[LINK]

---

## [Decision Letter · Decision Letter 2]

16 Sep 2020

Dear Dr. Rogne,

Thank you very much for re-submitting your manuscript "Body Mass Index and Risk of Dying from a Bloodstream Infection: A Mendelian Randomization Study of 56,000 Subjects from the HUNT Study With 23 Years Follow-Up" (PMEDICINE-D-19-04607R2) for review by PLOS Medicine.

I have discussed the paper with my colleagues and the academic editor and it was also seen again by two reviewers. I am pleased to say that provided the remaining editorial and production issues are dealt with we are planning to accept the paper for publication in the journal.

[LINK]

We look forward to receiving the revised manuscript by Sep 23 2020 11:59PM. 

Sincerely,

Thomas McBride, PhD

Senior Editor 

PLOS Medicine

plosmedicine.org

Academic Editor comments:

In response to Reviewer 4, point 2, related to the absence of winner’s curse, the authors have added to Discussion on p. 22: “As the instruments were selected from an independent sample, our MR analyses will tend to be conservative [42].” which is not correct, at least if I understand correctly what they mean for “conservative” – it would be indeed the opposite, i.e. presence of winner’s curse, to bias the results towards the null. The reply to point 5) is also not correct in relation to MR-Egger – the authors may check the findings of this pre-print paper, which addresses the specific issue of using 2-sample MR methods, including MR-Egger, in one-sample MR studies: https://www.biorxiv.org/content/10.1101/2020.05.07.082206v1.

Finally, I find it strange to read in the Abstract “Limitations of this study include reduced statistical power due to low sensitivity of blood culture tests”, given that the results were in fact all statistically significant – unless the authors refer to wide 95%CIs? The point about statistical power in Discussion is much clearer (“However, while blood culture tests are the gold standard to diagnose BSI, the sensitivity ranges between 10% and 50%, which may have reduced the statistical power in our study [44]. We overcame the issue of power by following ~56,000 subjects…”).

1- Please edit the data statement to read:

“Data from the HUNT Study used in research projects available upon request to the HUNT Data Access Committee (hunt@medisin.ntnu.no) to research groups who meet the data availability requirements (described here: http://www.ntnu.edu/hunt/data). 

BMI GWAS source data (Yengo et al, 2018) are available from the GIANT Consortium (https://portals.broadinstitute.org/collaboration/giant/index.php/GIANT_consortium_data_files)”

2- Please edit the title to: “Body Mass Index and Risk of Dying from a Bloodstream Infection: A Mendelian Randomization Study”

3- Abstract, line 109: please remove the sentence beginning “However, traditional observational studies…”

4- In the Abstract and throughout the manuscript, please include p-values alongside the 95% CIs for all comparisons.

5- The Abstract could note to investigate this in different populations as a potential limitation and/or future direction.

6- Abstract, line 126, please end the first sentence of the Conclusion with “... in this cohort” or similar.

7- Please spell out the abbreviation “HUNT” on the first mention in the Abstract.

8- Author Summary, line 161: Please begin the final point “In this cohort, higher genetically-predicted BMI was associated with…”

9- Did your study have a prospective protocol or analysis plan? Please state this (either way) early in the Methods section.

10- Line 199, perhaps “We sought to overcome the limitations of observational studies mentioned above by conducting…”

11- Thank you for following STROBE-MR. Please include the completed checklist as a supplemental file, and reference it (S1 Checklist). As this checklist is currently still a preprint (I believe), I think it’s best to also include the standard STROBE checklist (referenced as S2 Checklist). When completing each checklist, please use section and paragraph numbers, rather than page numbers.

12- In table 1, please note in the first column which categories are presented as mean (SD), median (IQR), or n (%).

13- Please note in each figure legend what the gray lines represent (95% CIs, I presume).

14- Results, line 328 and line 333, it seems that these results need to be presented as non-significant associations, as the confidence intervals cross 0.

15- Results, line 335-6, unless a statistical comparison between the two risks was done, please do not present this as a difference.

16- Discussion, line 382 (and all other instances)- please remove “prospective”.

17- Discussion, line 443, please include “To our knowledge…”

18- Line 469-470, you can remove “which greatly reduces confounding and reverse causation”

19- Please use the "Vancouver" style for reference formatting, and see our website for other reference guidelines https://journals.plos.org/plosmedicine/s/submission-guidelines#loc-references

Specifically, please include 6 author names before using “et al”.

20- For reference 25 (and any other preprints), please update the reference if the paper has now published (and check that the reference is still accurate). If the paper has not published yet, please include the date the preprint was accessed.

21- References throughout: Please remove spaces from the square brackets

22- Please remove competing interests from the end of the main text, and funding statement from the start.

Comments from Reviewers:

Reviewer #1: Dr. Rogne and colleagues have addressed my comments and the comments from other reviewers. The article has improved and I beleive it is now suitable for publication. 

I just found a typo error in Supplemental Figure 2, in which the Y-axis label says "Mortality" when it should be "Incidence". 

Congratulations to all the team for the great work.

Reviewer #4: I´m very satisfied with the authors´ replies. But they have still to address my comment 1, where I´m referring to the bias due to the binary outcome (see https://arxiv.org/pdf/1011.0595.pdf ) not due to confounding.

[LINK]

---

## [Editor Report · Decision Letter 3]

8 Oct 2020

Dear Dr Rogne, 

On behalf of my colleagues and the academic editor, Dr. Cosetta Minelli, I am delighted to inform you that your manuscript entitled "Body Mass Index and Risk of Dying from a Bloodstream Infection: A Mendelian Randomization Study" (PMEDICINE-D-19-04607R3) has been accepted for publication in PLOS Medicine. 

PRODUCTION PROCESS

Before publication you will see the copyedited word document (within 5 busines days) and a PDF proof shortly after that. The copyeditor will be in touch shortly before sending you the copyedited Word document. We will make some revisions at copyediting stage to conform to our general style, and for clarification. When you receive this version you should check and revise it very carefully, including figures, tables, references, and supporting information, because corrections at the next stage (proofs) will be strictly limited to (1) errors in author names or affiliations, (2) errors of scientific fact that would cause misunderstandings to readers, and (3) printer's (introduced) errors. Please return the copyedited file within 2 business days in order to ensure timely delivery of the PDF proof. 

If you are likely to be away when either this document or the proof is sent, please ensure we have contact information of a second person, as we will need you to respond quickly at each point. Given the disruptions resulting from the ongoing COVID-19 pandemic, there may be delays in the production process. We apologise in advance for any inconvenience caused and will do our best to minimize impact as far as possible.

PRESS

PROFILE INFORMATION

Thank you again for submitting the manuscript to PLOS Medicine. We look forward to publishing it. 

Best wishes, 

Thomas McBride, PhD

Senior Editor 

PLOS Medicine

plosmedicine.org